# Image Synthesis Pipeline for CNN-Based Sensing Systems

**DOI:** 10.3390/s22062080

**Published:** 2022-03-08

**Authors:** Vladimir Frolov, Boris Faizov, Vlad Shakhuro, Vadim Sanzharov, Anton Konushin, Vladimir Galaktionov, Alexey Voloboy

**Affiliations:** 1Keldysh Institute of Applied Math RAS, 125047 Moscow, Russia; vfrolov@graphics.cs.msu.ru (V.F.); vadim.sanzharov@graphics.cs.msu.ru (V.S.); vlgal@gin.keldysh.ru (V.G.); 2Faculty of Computational Mathematics and Cybernetics, Moscow State University, 119991 Moscow, Russia; boris.faizov@graphics.cs.msu.ru (B.F.); vlad.shakhuro@graphics.cs.msu.ru (V.S.); anton.konushin@graphics.cs.msu.ru (A.K.); 3Samsung AI Research Center, 125196 Moscow, Russia

**Keywords:** CNN-based sensors, synthetic training data, dataset augmentation, content creation pipeline

## Abstract

The rapid development of machine learning technologies in recent years has led to the emergence of CNN-based sensors or ML-enabled smart sensor systems, which are intensively used in medical analytics, unmanned driving of cars, Earth sensing, etc. In practice, the accuracy of CNN-based sensors is highly dependent on the quality of the training datasets. The preparation of such datasets faces two fundamental challenges: data quantity and data quality. In this paper, we propose an approach aimed to solve both of these problems and investigate its efficiency. Our solution improves training datasets and validates it in several different applications: object classification and detection, depth buffer reconstruction, panoptic segmentation. We present a pipeline for image dataset augmentation by synthesis with computer graphics and generative neural networks approaches. Our solution is well-controlled and allows us to generate datasets in a reproducible manner with the desired distribution of features which is essential to conduct specific experiments in computer vision. We developed a content creation pipeline targeted to create realistic image sequences with highly variable content. Our technique allows rendering of a single 3D object or 3D scene in a variety of ways, including changing of geometry, materials and lighting. By using synthetic data in training, we have improved the accuracy of CNN-based sensors compared to using only real-life data.

## 1. Introduction

The rapid development of machine learning (ML) technologies in recent years has led to the emergence of CNN-based sensors or ML-enabled smart sensor systems [1]. The advantages of such systems are most clearly manifested in the problems of medical analytics, unmanned driving of cars, UAV’s navigation, industrial robots, Earth sensing and photogrammetry, i.e., wherever computer vision is traditionally used. The efficiency of these systems highly depends on the training dataset.

Training of the modern neural network models in practice faces two fundamental problems: data quantity and data quality. The data quantity concerns availability of sufficient amounts of data for training and testing. The quality of data means how balanced the data is—are all the different classes that the model should recognize sufficiently represented, do we have precise marking for it or not, how much noise (of any nature) is present in the dataset? Real-life datasets collected by people usually suffer from both insufficient quality and insufficient quantity. In practice, the situation is even worse because computer vision (CV) researchers sometimes need to change input datasets for testing of certain hypotheses. Due to the inability to quickly obtain a new dataset, the CV researcher has to consider subsets of the existing dataset, which significantly limits their study.

Thus, synthetic (i.e., rendered using realistic image synthesis or other methods) datasets may be an option. The solution to the data quantity problem can be achieved through the use of algorithms for the procedural setting of the optical properties of materials and surfaces. This way, it is possible to quickly generate an almost unlimited number of training examples with any distribution of objects (and therefore classes) present in the generated images. The quality problem is actually more interesting. On the one hand, this is solved automatically: the renderer creates accurate per-pixel masks generating ground truth marking. On the other hand, it is actually not quite clear what quality actually means for CV.

Some researchers assume that quality is subjective realism to the human eye [2,3,4]. This is important to some extent, but rather wrong in general. Realistic rendering in comparison to simple or real-time techniques can improve accuracy of CNN-based methods. However, according to other studies [5,6] and our experience, the main problem is different: in practice, we measure the accuracy on specific datasets, and in this formulation of the problem, quality is understood as similarity to other examples of the dataset, finally used to measure the quality of the CNN-based algorithms. This in itself significantly shifts the priorities of the developed approaches and software for our solution.

The purpose of our work is to propose a controllable and customizable way for inserting the virtual objects and variating their distribution and appearance in a real dataset. This allows us to both improve the accuracy of CNN-based methods and to conduct experiments to study the influence of various factors on the accuracy. In most cases, it is impossible to a priori determine the initial distribution of the features we are interesting in. It depends on the original dataset. The mechanism we propose allows us to investigate its impact by adding objects with certain features, and in some cases, the impact of the particular feature may be negligible. Unlike existing works, we propose a general dataset augmentation pipeline that we have tested on many different scenarios and datasets. Therefore, our contribution is in a successfully validated software solution that allows one to augment and expand training datasets in a controlled way. Some of the results of our work are shown in Figure 1.

## 2. Related Work

It is very important for the generated images that the embedded synthetic objects look realistic in the scene. Because neural networks can be trained to recognize synthetic objects by the artifacts that appear when they are added. The creation of high-quality synthetic samples will reduce the cost of data collection since the amount of real-life data can be reduced. Additionally, mistakes are often possible when data is manually marked up by people. Synthetic data allows us to get perfectly accurate data markup.

There are a huge number of successful applications of synthetic datasets for computer vision. For us, first of all, it is interesting how the dataset was synthesized and how 3D content was obtained in these works, how the full pipeline was built and what were the advantages and disadvantages of the selected methods.

### 2.1. Using Existing 3D Content

First, Movshovit et al. [2] demonstrated the importance of the realistic rendering usage in the problem of view point estimation. In their work, authors used a database of the 91 3D CAD models obtained from the sites doschdesign.com (accessed on 1 December 2021) and turbosquid.com (accessed on 1 December 2021). These models were rendered with randomized direct lighting and random real photos in the background. We believe this work was not measuring the influence of realism itself, but rather the effect of increasing the dataset variability, which has an effect with even simple lighting models to some extent. The reason is obvious: 3D models downloaded from the Internet resources are in most cases not ready for photorealistic rendering due to the lack of textures, unknown materials models (which are simply incorrectly exported to the current format or not supported by the current rendering system) and, finally, just human errors made when preparing a 3D model for an aggregator site.

Zhang et al. [7] reports about using 500 K images from 45 K scenes for training of semantic segmentation and normal/depth estimation problems. The 3D content was obtained from the SUNCG dataset [8]. There are many problems with this approach, even apart of the well-known legal problems with the SUNCG dataset. Since originally the SUNCG 3D models were assembled in the Planner5D program (which is trivial compared to such content creation tools like Blender or 3ds Max), they do not contain realistic materials or lighting settings. They are limited to textures for Lambert reflection only. This content is not sufficient for photo-realistic rendering. No wonder that the other works where the SUNCG was used [9,10,11] could only achieve basic level of realism.

### 2.2. Augmented Reality

Preparation of 3D content is an expensive and time-consuming task. Therefore, in practice it is worth to consider any possibility for reducing man-hours spent on this. Augmented reality is a very advanced approach here. Even [2] actually used it when randomizing backgrounds. The authors of [12] went further, applying augmentation of rendered images to the KITTI dataset [13]. Applying of the image-based lighting technique [14] to individual car models in combination with augmenting real photos gives a cheap and fast result: both rendering computation complexity and content preparation are greatly simplified [12].

The main problem of image-based lighting methods is that they need the real high dynamic range to be present in an HDR image. For example, on a sunny day, we have a dynamic range of 5–6 orders of magnitude. This is often not so [14]. Lighting prediction methods [15,16,17,18] can help here. They have become a quite common task for mixed reality systems. We would like to highlight Pandey et al.’s work [19], where portrait relighting and background replacement were suggested. The method works well on faces, but unfortunately requires a training dataset that is hard to obtain for every domain of application.

Another disadvantage of augmented reality is the task of automatic, correct placing of 3D models inside some reconstructed representation of the scene. In some cases it can be easily provided (for example, ref. [12] used segmentation methods to reconstruct a road plane), but in general it is quite difficult (for example, for interior photographs). Thus, the approximate geometry, lighting and camera parameters including tone mapping should be reconstructed in some way and this reconstruction itself is a significant problem. Nevertheless, the main advantage of augmented reality is that it can work directly in the image domain of CV researcher interest expanding existing datasets with new examples. And thus, Domain Adaptation methods [6] are not needed which seems to be mandatory in practice to achieve high quality by other syntheses ways.

### 2.3. Procedural and Simulation Approaches

Procedural modeling is expensive, time consuming and is usually limited to some specific task (such as modeling of dirt or fire) and sometimes to a specific rendering algorithm. However, once created, procedural models are highly variable (an infinite number of examples), provide excellent rendering quality (often with infinite resolution details) and are traditionally used in many cinema content creation pipelines. Tsirikoglou et al. [3] used a procedural modeling approach to generate cities and showed an improvement for neural networks. Buildings, roads and city plan were procedural, while other models like cars, pedestrians, vegetation, and bicycles were randomly sampled from a prepared database. Some car parameters were also randomized: type, count, placement, and color.

Approaches based on procedural modeling with machine learning look promising [20,21], but here we run into a chicken-and-egg problem: we need to train these models first. Besides, these approaches are rather new and have not yet been proven in practice of general rendering tasks.

Hodan et al. [4] achieved high quality by using the existing film production content creation pipeline with Autodesk Maya and Arnold rendering system, high quality 3D models and physics simulation as the main randomizing tool. Obviously, this approach suffers from the main disadvantage of current film production pipelines: high cost and high labor input. Hodan et al. [4] used only six scenes with 30 different objects. This is fundamentally different from previous approaches. Other disadvantages of Hodan’s approach include the use of non-freely available tools Maya and Arnold (which limits the adoption of this work) and slow rendering reported by authors. They needed 15 to 720 seconds per image (depending on quality) on a 16-core Xeon CPU with 112 GB of RAM. Therefore, reasonably fast creation of a dataset is possible only if there are significant computational resources. It is important to note that authors reported that accurate modeling of the scene context was more significant (+16% for CNN precision) in comparison to accurate light transport simulation (+6% for CNN precision).

The game engines can also be used for dataset generation [5,22,23]. The choice between a game engine or a film production rendering system should be made primarily based on how easy it is to model the target real-world situation.

### 2.4. Content Creation Pipeline

Unlike other works mentioned earlier, a specialized content creation pipeline is proposed in [24] and, as was stated by the authors precisely, their goal was to design an open-source and universal pipeline. So, it is very close to our work. Sampler modules provide randomization capabilities, which are the most interesting part of the pipeline. Sampler modules can generate positions for object placement (cameras, lights, 3D models) with various distributions and constraints such as proximity checks. So, for example, the object positions can be generated on a spherical surface, but the objects will not be too close (for example, the cameras will not look straight at the wall) and/or will not collide with each other. Sampler modules can also select objects based on user-defined conditions and manipulate their properties (for example, enable physics simulation for 3D models [25]). The MaterialManipulator and MaterialRandomizer modules have been implemented to produce variation in appearance.

The main problem of this work is that, so far, it has not yet been applied to any specific training task unlike many previously mentioned works. So, although it looks very good on paper and in the examples provided by the authors, its performance in real-life applications for CNN training is not yet known. As mentioned earlier, the authors intended their work to be a universal pipeline that could be adapted and used by CV researchers. Therefore, real evaluation of their work can only be made when its applications are published. We believe that the [24] approach is generally correct. However, having experience of similar pipeline creation and its application to various computer vision tasks, we would discuss several modifications that need to be made in order to put the Denninger approach into practice.

Another interesting idea is to create datasets from 3D scans [26]. This approach has advantages over fully generated images (and the combined approaches discussed in the next subsection) because the resulting datasets are more natural in terms of both image quality and semantic information (especially for interiors). However, this usually requires specific hardware and software. Besides, not everything we need can be easily found in reality (for example, new car models that have not yet been constructed). Therefore, computer graphics or combined approaches are exactly what we need for such cases.

### 2.5. Applying Generative Neural Networks, Combined Approaches

The process of collecting and labeling training samples for solving computer vision and machine learning problems is a very costly procedure. The quality and quantity of data determines the accuracy that approaches can achieve to solve the corresponding problems. Data augmentation is actively used in the process of training algorithms. High-quality synthetic sampling can help solve the problem of insufficient data.

Modern methods for generating synthetic images are based on Generative Adversarial Networks [27]. In short, two networks are trained in such methods: a generator and a discriminator. With the help of alternative training, they learn how to create realistic images and distinguish them from real ones. These approaches are now being actively developed.

Embedding random objects in datasets in the paper [28] was proposed by creating compositions of some background images with object images. A special neural network called “Spatial Transformer Network” [29] in it predicted the parameters of the affine transformation that needs to be applied to the image of the object so that it looks most natural against the corresponding background. The discriminator is trained to distinguish real images from synthetic ones. As a result, the authors managed to improve the quality of the Faster-RCNN detector [30] trained on the COCO dataset [31].

To build realistic compositions, methods are needed not only to train how to find the right place to embed a synthetic image, but also to adjust the lighting of the embedded object to the lighting of the background on which it is located. It is also necessary to get rid of artifacts that appear when embedding objects. Therefore, the authors of paper [32] proposed a different approach, also based on generative adversarial neural networks. The authors proposed an additional full-convolutional neural network, an autoencoder, that takes a composition of the background and new object images as input, and predicts an improved original image at the output. It is assumed that this part of the architecture will remove artifacts on the border of the background and the embedded object. Furthermore, additional regularizing terms have been added to the loss function so that incorrect affine transformations are not obtained.

Neural networks can be used not only to determine the location of a new object, but also to increase the realism of already embedded objects. This is a domain adaptation problem, and it can be successfully solved with the help of generative adversarial networks. An example of such an architecture is CycleGAN [33]. Two generator models and two discriminator models are used here. They are trained simultaneously. One generator takes images from the first domain and generates images to the second domain. Vice versa, another generator takes images from the second domain and outputs images for the first domain. The discriminator models determine the plausibility of the generated images. An idea of cycle consistency is used in the CycleGAN. This means that the output image of the second generator should match the original image if the image output by the first generator was used as input to the second generator.

A new combined approach called Sim2SG has been developed recently [23]. It matches synthetic images to the real data distribution. In this approach, synthetic data is generated in a loop consisting of two alternating stages: synthesis and analysis. During the synthesis stage, scene graphs are inferred from real data. Then, synthetic data are created to match the distribution of real data. Thus the content label gap is eliminated. During analysis stage, a scene graph prediction network is trained using synthetic data. To bridge the gap in appearance and content prediction, Gradient Reversal Layers (GRLs) are used.

MetaSim (and Meta-Sim2) [34] is designed to automatically tune parameters based on a target collection of real images in an unsupervised way and aims to learn the scene structure in addition to the parameters (Meta-Sim2). The goal of this approach is to automatically capture statistics of objects and parameters of procedural models, such as their frequency, on real images. Therefore, the generated scenes have close-to-real semantics of object placement, their appearance, etc. The idea is to build a Grammar-based model which parameters could then be tuned automatically.

### 2.6. Summary of Existing Approaches and Motivation for the New Method

Firstly, existing approaches to randomization of material parameters are rather limited: existing works are limited to randomization of color or material type [2,4,5,12,24]. Changing other parameters of material models, as well as using procedural texturing, has a great potential to increase the visual variety of synthesized images, but have not yet been studied.

Secondly, the existing solutions do not address the issue of controllability and reproducibility of random scene generation, which is very important in the context of generating a dataset with the desired distributions of objects and their characteristics. In order to check the influence of a certain factor on the accuracy of the CNN-based methods, we must be able to generate “almost the same” datasets, which at the same time differ in a certain way. For example, if we want to carefully check the impact of dirt modeling (on cars) to the accuracy, we must reproduce exactly the same position, colors, etc., of rendered cars with and without dirt. This reproducibility is essential because it helps to reduce the generator error in assessing the accuracy of the neural network. More precicely, we do not introduce additional error caused by a difference in other, not currently investigated features, neither by randomness itself. Indeed, different positions, colors and lighting introduce additional error that we have to measure by training the CNN multiple times. This approach would be impractical for CV researchers due to the prohibitively expensive training process.

Recent works try to automatically fit the desired distribution of synthetic objects in some dataset [23,34] which, in our opinion, is significant progress in dataset augmentation. These works, however, are not intended for customization of CV experiments, which is important for detailed investigation of CNN properties (like what if we change the appearance of this feature, or the type of car, or the lighting, etc.). This is also a problem for scanning based approaches [26].

Finally, most of the existing works are limited to specific applications, which do not allow generalizing the proposed approaches and draw general conclusions.

Thus, the purpose of our work is to generalize the approaches of using synthetic images to improve the accuracy of the CNN-based computer vision algorithms by creating a general and customizable augmentation pipeline. It should allow us to conduct exactly the experiments we want and test any specific hypotheses on the generated datasets. Unlike existing works, we demonstrate the successful application of the developed pipeline in various computer vision CNN-based applications, which allows us to draw some general conclusions.

## 3. Proposed Approach

Our system has several key components:software for 3D modeling (Figure 2, “3d modeling software” box);software for rendering (Figure 2, large rounded box at bottom-right);database of 3D content (Figure 2, large rounded box at bottom-left);tools for post-processing of images (Figure 2, “Image Processing” box);scene generation system (Figure 2, “Input scenario”, “Randomization tools”).

We describe them further in this section.

Input scenario script defines the settings for the entire generation procedure, i.e., for all other components of the system: classes of 3D models, types of lighting (indoor or outdoor, day or night, etc.), random generation settings (which parameters are randomized and with what distribution), the output of the rendering system (segmentation masks, binary object masks, normals, depth map, etc.), and post-processing of the image that needs to be done after rendering.

The database contains the main resources for generating scenes:3D models with setting up material tags (the mechanism of tags will be discussed later);Materials—finalized (i.e., ready for rendering) materials with their tags;Textures—set of texture images and normal maps for use in materials;Environment maps are used for image-based lighting, representing different lighting conditions;Content metadata—information that is used by randomization scripts to select 3D models, materials and other resources based on the generation script and internal information (for example, material tags and goals).

Artist Tools are specially designed tools as an extension to 3D modeling software that allow us to set constraints for the randomization process during the preparation of 3D models and materials.

Randomization tools generate the requested number of scene descriptions according to the input scenario. The generated scene description is intended for direct use by the rendering system.

Rendering system. In this work we used Hydra Renderer [35], which implements modern photorealistic rendering algorithms and provides high performance (which is especially important since a large number of images must be generated). The Hydra Renderer uses a well-structured XML scene representation that is easy to create and edit with scripts and thus define the scene using rendering tools. This description also includes information about which procedural effects should be used and what their input parameters (if any) are. Hydra Renderer supports custom extensions for procedural textures and we make extensive use of this functionality.

Image processing tools adjust output images to provide additional data or combine them with input images if an augmented reality generation approach is used. An example of additional data generated in this step is bounding boxes for individual objects. We want the resulting neural network model to show an improvement in the quality of work, for example, according to the classification accuracy criterion, on a certain dataset of the real world. Thus, it is necessary to be able to generate a dataset that would have characteristics that, on the one hand, would be similar to this real world dataset, and on the other hand, would complement it. The similarity implies, among other things, the characteristics of the images themselves: the presence of optical distortions, the parameters due to the sensors used in practice to obtain images. For these purposes, the image processing component may include, for example, adding chromatic aberrations, a distortion effect, blurring, transforming and deforming the image, adding noise, etc., depending on the images in the underlying real-world dataset that is being worked with.

### 3.1. Randomization Tools, Scene Sampling

A significant difference between our work and existing approaches is the reproducibility of scene generation results, which allows generation of the same desired features of a scene for the same input setup. In fact, this turns dataset generation into a sampling problem. Having an input vector $\left(x_{0},x_{1},\ldots x_{n}\right)$ of random numbers from 0 to 1, we assign a specific value to each random variable during randomization script development.

Let us consider this approach on a concrete example, Figure 3. If we would like to render cars in an HDR environment, like [2,12] did for 3D model pose/camera parameters estimation [2] or training car detector [12], we could:Assign a pair $\left(x_{0},x_{1}\right)$ to rotate the 3D car model (additionally, you need to set the allowable angle intervals, for example, from 0 to 70 degrees for vertical angle and from 0 to 360 degrees horizontally).Assign $x_{2}$ to select a car body color from palette (to limit desired colors).Assign $x_{3}$ to select the environment.Assign $x_{4}$ to rotate the environment around the vertical axis (Y).Assign $x_{5}$ to select the 3D car model.

We call this process scene sampling. All parameters are mapped into the interval [0,1]. Thus, by sampling of the 6D vector $\left(x_{0}\ldots x_{5}\right)$ we obtain images for the input samples, which gives us important properties and thus powerful capabilities for further experiments:Applying quasi-random sequences (we used the Sobol sequence), we kill two birds with one stone: (1) Running the Sobol sequence from the beginning for each 3D model (i.e., generate $x_{0}\ldots x_{4}$ with a quasi random approach, but select 3D models one by one instead of sampling $x_{5}$) allows us to generate the same camera positions for all 3D models, which is important for training pose estimation algorithms. (2) The Sobol sequence gives a good uniform distribution in 4D–6D space. So, the generated samples will also be evenly distributed among all desired parameters: car colors, camera rotation and lighting conditions (which are set from the environment map).Applying a non-uniform distribution for $x_{5}$ allows us to select more cars of the desired types and, applying a tabular distribution for $x_{2}$, we can use more cars of specified (for example, black and grey) colors.By setting different values and distributions for the training and validation datasets, we can easily check our hypothesis of interest. For example, we can generate a training dataset with camera rotation for only 25, 50 and 70 degrees, but then check if this is enough for the model to identify 10, 30 and 60 degree rotations in our test dataset.When we move from the current frame to the next one, we would like to localize the scene changes. In this case, the rendering system does not often have to perform expensive operations to load content from disk into GPU memory. So, we efficiently apply the caching technique. In our experiments, we can automatically get the desired caching for the rendering system by clustering and reordering multidimensional input points. For example, we can group all input samples by the pair ($x_{3}$, $x_{5}$). This way, we consistently render scenes with the same 3D model and the same environment map.

We would like to make a special remark on the last point. It can be assumed that this optimization is not essential and can be omitted or achieved in other ways. For example, if we need to reproduce [2]’s work, the researcher can manually modify the script logic to optimize resource usage: load a specific environment map and a specific 3D model, generate samples for this environment map, then modify the 3D model of a car, but use the same environment map. In our opinion, this approach is very narrow and impractical because it is difficult to generalize it for other generation scenarios. Thus, we believe the following reasons support our position in practice:When a researcher writes a randomization script, they most likely do not know the details about the rendering system. So, effecient cache usage should be applied automatically without polluting the randomizer script logic with such optimization issues.Some specific procedural approaches can be done directly on the GPU (this is what we actually did for procedural textures). However, it is almost impossible to avoid costly interaction with disk or CPU simulation tools (for example, vector displacement, physics based animation or simulation) because realistic simulation/modeling often has to rely on a lot of existing CPU-based products that could work for minutes or even hours (although it gives realistic results). So caching is important.Our optimization is critical for systems based on real-time rendering algorithms if they directly feed their result to the neural network on the GPU. Even in our case (we use a “offline” GPU path tracing renderer), we gain acceleration from 1.5 to 3 times for this case because the input HDR images are high resolution 8 K × 4 K (which is the normal size for most of existing HDR environment assets), and it is simply impossible to load such a texture from disk quickly enough.

Thus, in our approach, resource usage is optimized automatically by simply reordering and grouping input samples by their coordinates. This is general, usable and can be carefully tuned for any generation scenario.

### 3.2. Artist Tool

In practice, writing randomization scripts turns out to be a rather laborious task, because after writing the first version, you need to run the script, check the generated images (at least several dozen images selectively), change the script, if something went wrong, run the script again, etc. To improve this process, some of the randomization work has been implemented as an artist tool integrated with 3D modeling software. The main task solved by this tool is to ensure that random distributions of the material parameters are adjusted already at the stage of preparing 3D models and materials so that an acceptable result from the point of view of realism is obtained during generation.

Unlike traditional approaches to the process of creating 3D models (in 3ds Max, Blender or other software), our solution allows the artist to create not one, but a whole set of 3D models with a given distribution of the optical surface properties, i.e., the parameters of the material model and procedural textures (which are actually also material properties). Figure 4 shows a preview of the randomization results in our artist tool.

#### Material Tags

In order for the assignment of random materials to 3D models to give an adequate result from a human point of view, two features must be taken into account.

First, two materials of the same nature, such as wood, cannot really be interchangeable in all cases. Take, for example, a wooden pencil and a wooden floor. Both of these objects have a wood material, but have different wood microstructures and, most importantly, different textures and texture matrices (in particular, the components responsible for scaling and rotating the texture). Therefore, assigning a floor material to a pencil looks not only strange, but in fact completely wrong. When the camera zooms in on the pencil, we will see small copies of, for example, a parquet board on it. And if the virtual observer is located far away, then, most likely, we will just get some kind of unknown color (average color over all pixels of the texture).

Second, there are many materials that can only be applied to certain types of objects: TV screen or computer monitor material can be displayed on an advertising poster and vice versa, but cannot be displayed on a cup or any other furniture, and the cup materials can have a texture wrapped around the cup 3D model. Images for such textures are horizontally elongated rectangles (usually 2 × 1), and applying this texture to many other objects gives a completely unrealistic appearance with a stretched image.

Based on these features, we divided all randomized materials into two different categories: universal (also called tagged) and special (also called targeted). Generic content uses a list of tags. The tags determine what types of output materials can replace this material when generating a scene.

Special materials mean that such materials can only appear on certain types of objects, called targets. For example, a TV screen, a traffic sign, a book cover, car tires. Due to the specific texture coordinates and the actual content of the texture, it is not possible for these cases to replace these materials with some other materials and vice vesa to apply them to other objects. Therefore, the target material can only be replaced by another target material for the same purpose.

The work on preparing 3D models and materials using the developed tools is divided into two stages: filling the material database and filling the object database. At the first stage, customized materials are created and tags or targets are specified for them. A single material can be of multiple types if it can be combined with other materials or used as part of a complex material model. At the second stage, we assign dummy materials to 3D models from the database and specify only tags or targets for these materials. At this point, we can run a randomizer check (Figure 4), change the tags/target, or return to the first step if we need to create additional materials.

### 3.3. Procedural Textures

As mentioned earlier, procedural textures were extensively used in the developed approach. The main purpose of using procedural textures was to add additional detail to rendered 3D models to create more realistic and valuable images. As part of the work reported also in [36], procedural textures were implemented that imitate such effects as dirt, rust, scratches, icing (Figure 5).

All of these textures are parameterized, which makes it easy to change the appearance of the texture at the stage of scene generation and rendering. Most of these procedural effects are based on noise functions, so noise function parameters such as amplitude, frequency, and constancy (or coefficients applied to these parameters in some way) can be used as scene parameters and can be randomized. In addition, one of the parameters of procedural textures is the relative size of the object. This parameter allows you to control how the procedural effect spreads across the surface of an object. For example, a procedural dirt texture can be set to apply only to the bottom of the model. This kind of flexibility is not possible with conventional image textures.

It is important to note that using a procedural approach to texturing is critical in some cases because 3D models may not have the valid texture coordinates. This is a fairly common case, even with high quality open-source 3D models. Procedural textures can take world (or local) coordinates of a surface point as input that can be used to render textures. In addition, even with texture coordinates on the model, there is no guarantee that textures such as dirt or rust will look correct on different 3D models due to the scaling factor of these models when converting from local to world space. In such cases, using procedural 3D textures that depend on the position in the world space is the only option.

## 4. Experimental Evaluation

The purpose of our experiments is primarily to evaluate the influence of various factors to the accuracy of the CNN:How valuable it is to be able to realistically render the object of interest itself;How valuable it is to implement the correct placement of objects (when augmenting a real photo);How valuable post processing styles are;How valuable is the application of the computer graphics approach in comparison to the synthesis via neural networks.

Thus, these questions determine the nature of our experiments.

### 4.1. Cars and Road Signs

The first experiment implemented according to the scheme using augmented reality is in many respects similar to [12], where for the available images from the dataset [13] trajectories were semi-automatically marked on road sections (Figure 6), after which the positions on these trajectories were randomly sampled.

To prepare data for car augmentation, we need to convert the camera calibration matrix to a OpenGL frustum matrix and to mark the trajectories on the frames. Given the intrinsic camera calibration matrix K and the extrinsic matrix [R|t], we do the following:Set [R|t] as the worldview matrix for OpenGL.Convert intrinsic matrix *K* to a frustum matrix using the formulas:$$K=\left(\begin{array}{ccc} \alpha & s & -x_{0}\\ 0 & \beta & -y_{0}\\ 0 & 0 & -1 \end{array}\right)$$
$$left=-\frac{near}{\alpha *x_{0}},right=\frac{near}{\alpha *x_{0}}$$
$$bottom=-\frac{near}{\beta *y_{0}},top=\frac{near}{\beta *y_{0}}$$
$$P=\left(\begin{array}{cccc} \frac{2near}{right-left} & 0 & \frac{right+left}{right-left} & 0\\ 0 & \frac{2near}{top-bottom} & \frac{top+bottom}{top-bottom} & 0\\ 0 & 0 & -\frac{far+near}{far-near} & \frac{-2far*near}{far-near}\\ 0 & 0 & -1 & 0 \end{array}\right)$$

Matrix *P* is called a projection (frustum) matrix in OpenGL.

The trajectories on the images were marked as piecewise linear functions (Figure 6). After marking the pixels, project these pixels using camera calibration matrix. We assume that the ground lies on the XOZ plane. This assumption gives a unique projection of the pixels to the world coordinates. Some examples of generated images are presented in Figure 7 and Figure 8.

### 4.2. Augmenting Interior Images

The second approach we tested also used augmented reality because our primary goal was to embed new objects into an existing dataset. In this case, the objects were embedded using a depth map. Two different options were used. In both versions, a point cloud was generated from the depth map. Then, it was processed: the statistical outliers were removed, the level of detail was reduced using a voxel grid, and normals were restored.

#### 4.2.1. Augmenting New York (NYUv2) Dataset

In the first variant, planes were additionally extracted from the point cloud. Then, using the approach described in the Section 3.1, one of the planes was selected with one random number, on which a point was then chosen for embedding the 3D model, depending on whether the plane is vertical or horizontal. Figure 9 and Figure 10 show examples.

#### 4.2.2. Augmenting DISCOMAN Dataset

In the second variant (DISCOMAN dataset [10]), we know the position of the camera and the floor from the dataset itself. For this case, we developed an occupancy map reconstruction algorithm to place objects on the floor behind (or in front of) the real objects in the scene. The occupancy map is a binary image, the purpose of which is to show the free space in the scene (where synthetic objects can be inserted) and the occupied parts of the scene (Figure 11). Examples of our augmentation can be found in Figure 12, Figure 13, Figure 14 and Figure 15.

To obtain 3D human models, we used SURREAL [39]. We reorganized the existing 3D model generator in accordance with the general approach of our work in such a way as to control the shape of the human model and vary the base mesh between “thin/full”, “low/high”, etc., feeding a vector of random numbers as input.

### 4.3. Fit the Lighting with GAN Post-Processing

One of the serious problems of augmentation is that the lighting conditions of the rendered and real objects in the image may not match. This is because we do not really know the lighting settings for the augmented image. As we already mentioned, the lighting prediction methods [15,16,17,18] can help here. However, their application has a limitation: different methods should be used for different datasets (for example, indoor and outdoor scenes). So, these approaches are not automatic and must be tweaked significantly for each data set.

Our goal was to elaborate an automatic method that allows us to fit the image of a rendered object as if it was lit by the same unknown lighting conditions that are present in the source image, even if the original lighting conditions on the render were different. For this purpose we trained CycleGAN on a fully synthetic dataset where we rendered various 3D models in an HDR environment (Figure 16).

We took as a basis the modified CycleGAN [33] architecture described in [40]. In our task, its architecture consists of two encoders (one for each class of pictures) and two decoders (for each class of pictures). One of the features of the architecture is the Relativistic loss that is used for both the discriminator and the generator. For our experiments, we used only one discriminator of RGB images. The general scheme of the method is shown in Figure 17.

Then, we tried several modifications of this method. In the task of embedding synthetic objects into the background, we know in advance which part of the image has been changed because we have masks for the embedded objects. Therefore, we would like to transform only embedded objects and not change the background. As a first modification, we decided to apply not only a three-channel image to the generator and discriminator, but also a mask of embedded objects as the fourth channel. We also added SmoothL1Loss, which applied only to the background and penalized the GAN for changing it. Object masks have been blurred around the borders so that there are fewer artifacts around them. Thanks to this blurring, SmoothL1Loss penalized background changes on the borders of objects less than the rest of the background and provided a smooth object border. As a second modification, we tried to cut the background from the original image and paste it into the processed image. This method performed slightly worse than the previous one without background cutting.

We also trained the neural network for the panoptic segmentation task on different datasets and then evaluated the models on the test set of NYUv2 and DISCOMAN datasets. We calculated the panoptic segmentation metrics. The results for “aug-gan” in the Section 5.4 and Section 5.5 demonstrate an improvement in quality.

## 5. Validation and Results

We proposed a pipeline for embedding synthetic objects into scenes. We conducted an experimental evaluation of the proposed approach in 4 different scenarios. To do this, we trained neural network models on synthetic data for (1) classification, (2) detection, (3) depth map assessing, and (4) panoptic scene segmentation. We then used the GAN model to improve the visual realism of the resulting synthetic scene and studied its influence on the accuracy of the investigated tasks.

### 5.1. Results on Traffic Sign Classification (RTSD Dataset)

Using our pipeline for traffic signs, we were able to increase the accuracy of the classifier for the WideResNet [41] **rare** traffic signs from **0%** to **70**–**76%** [42]. At the same time, in order to preserve accuracy on the **frequent** traffic signs, we had to use a hybrid approach where the rendered images were further styled with a neural network. Results are present in Table 1, where Accuracy and Recall are standard metrics.

RTSD—the classifier is trained on RTSD dataset only.CGI—samples were obtained by rendering three-dimensional models of traffic signs on pillars in real road images.CGI-GAN—traffic signs were transformed from the CGI collection to better ones using CycleGAN [33].Styled—a hybrid approach where the rendered images were further styled with a neural network.

### 5.2. Results on Traffic Sign Detection (RTSD Dataset)

For traffic sign detector, we have found that modeling the context in which traffic signs occur is more important than modeling the traffic signs themselves [43]. We conducted experiments with basic methods for inserting a sign image onto the original image of an unsigned road (the basic methods do not take into account the context of a traffic sign):**CGI**—samples were obtained by rendering three-dimensional models of traffic signs on pillars in real road images.**Inpaint**—this is a simple synthetic data for the detector, in which an icon of a traffic sign is drawn in the image without any processing.

Next, we also explored various approaches for inserting generated traffic signs into the original image.

**Pasted**—In this approach we trained together neural networks for inpainting and processing of embedded traffic sign [43].**Styled**—In this approach we used style transfer via neural networks [44] to make inserted image more close to the original background image.**NN-additional**—In this approach we trained a neural network for account object context and thus find better places to insert generated traffic signs [43].

Result can be found in Table 2.

It can be seen that without proper placement of road sighs (RTSD + NN-additional row in Table 2) with the help of a neural network we were not able to improve the detection quality of **frequent classes** using any synthesized data. This is 89.25 for frequent classes when training only on real data. The best quality using synthetic samples without neural network placing is achieved on Styled and is equal to 89.13. At the same time, for **rare signs** AUC (Area Under the Curve) increases from 85.86 to 86.78 [43] with the hybrid ‘Styled’ approach.

### 5.3. Results on KITTI Car Detection Dataset

Here, we examine the quality of cars rendered using the proposed pipeline to augment training samples for a car detection task. We use the KITTI-15 dataset [12] for experimental evaluation. It consists of training and test parts, each contains 200 frames. Since there is no labeling for the test set, we manually labeled the bounding boxes for the test set. The training set was considered as a set of backgrounds into which rendered cars were inserted. We augment each image of training set 20 times with 2–5 random cars. As a result, we have 4000 frames in the augmented training set.

The purpose of this experiment is to evaluate the contribution of various generator/renderer features, as well as accurate modeling of the characteristics of the dataset in our samples, to the quality of detector training. The following datasets were prepared for this experiment:**Aug-1**—all features are enabled, uniform distribution of all generator parameters is used: car body colors, cars types, etc.**Aug-2**—wide range of colors including variegated. This setup shows the impact of accurate modeling of the distribution of car colors in the generated sample.**Aug-3**—disabled features: transparent windows, so there are only dark glasses.**Aug-4**—disabled features: dirt, smudges, rust, dents.**Aug-5**—disabled features: shadow.**Aug-6**—disabled features: unusual appearance (police, ambulance, etc.).**Aug-7**—wide range of rotation of inserted cars. This setup shows the impact of accurate modeling of car orientation.**Aug-8**—one car of each type is inserted into one image. This setup shows the impact of uniformity in the distribution of rendered car types.**Aug-9**—only hatchbacks are inserted. This setup shows the impact of accurate modeling the distribution of car types.

In all experiments, Faster-RCNN [30] with a ResNet-50 FPN backbone detector was used to localize vehicles. Following [12], we used only 2000 iterations of SGD with batch size 4. 500 out of 2000 are warm-up iterations with a linear increasing of learning rate. The base learning rate is 0.005.

We use mAP metric (mean areas under the curve for various IoU thresholds) to evaluate the quality of the detector. The resulting mAPs are shown in the Table 3.

It is clearly seen that augmentation of frames with rendered cars improves the quality of the detector (mAP metric increased from 40.3% to 43.26%). During the experiment, it was found that KITTI augmentation via the proposed pipeline improves the training of the detector when tested on both datasets, Cityscapes and KITTI. Moreover, all rendering features are important, as can be seen from Table 3. The variety of inserted models is not very important, augmentation with one type of car is close in AP to full augmentation.

However, the complexity of this dataset (KITTI) is not enough to use it for further testing on a complex and variable collection like NEXET or BDD100k. Models trained in this way do not work on such datasets at all.

### 5.4. Depth Estimation

Our first test of the image augmentation pipeline is the depth estimation task. We use the single-view depth estimation method from [46]. This method is based on the encoder–decoder architecture. The input RGB image is encoded into a feature vector using the DenseNet-169 network pre-trained on ImageNet. This vector is then fed into a successive series of upsampling layers, to construct the final depth map. These upsampling layers and their associated skip-connections form their decoder. Their loss function is a weighted sum of three loss functions: mean l1-error on the depth map, smoothness (l1-loss over the image gradient), structure similarity loss.

The implementation of a method proposed by the authors uses inverted depth maps. That is, they are transformed with the form C/y (C is a constant), where y is the original depth map. The synthetic images and depth maps generated by us have a different scale than the real ones. Because of this transformation (C/y), we could not make the synthetic depth maps look like real ones for the neural network. Therefore, we changed the data generator, now it uses the original depth maps. Synthetic depth maps are multiplied by a factor of 25. To achieve the same performance as in the original paper, we also changed the weights in the loss function. Now, the component responsible for the depth map smoothness is summed up with a weight of 0.3.

We used six standard metrics [47] to evaluate the accuracy of our model:Average relative error (rel): $\frac{1}{n}\sum_{p}^{n}\frac{\left|y_{p}-{\hat{y}}_{p}\right|}{y}$.Root mean squared error (rms): $\sqrt{\frac{1}{n}\sum_{p}^{n}{\left(y_{p}-{\hat{y}}_{p}\right)}^{2})}$.Average ($log_{10}$) error: $\frac{1}{n}\sum_{p}^{n}\left|\log_{10}\left(y_{p}\right)-\log_{10}\left({\hat{y}}_{p}\right)\right|$.Threshold accuracy ($a_{i}$): % of $y_{p}$ s.t. $\max(\frac{y_{p}}{{\hat{y}}_{p}},\frac{{\hat{y}}_{p}}{y_{p}})=a<thr$ for thresholds:$1.25,1.25^{2},1.25^{3}$.
where $y_{p}$ is a pixel in depth image *y*, ${\hat{y}}_{p}$ is a pixel in the predicted depth image $\hat{y}$, and *n* is the total number of pixels.

#### 5.4.1. Results on NYUv2 Dataset

We used several collections of the same dataset. Let us introduce the notation:small—a small collection of 795 training data with semantic segmentation markup.aug—the first collection of augmented data generated from “small”, 7950 pictures total.aug-gan—set of synthetic images from “aug” after post-processing with GAN.

First, we trained the neural network for the depth estimation task on different datasets and then evaluated the model on the NYUv2 test set. The results are in Table 4.

It can be seen that our pipeline improves the accuracy of depth estimation on the NYUv2 dataset. RMS decreased from 0.5807 to 0.5706. However, augmentation of rendered images with GANs did not give us a stable accuracy improvement over augmentation with a simple blur filter.

#### 5.4.2. Results on DISCOMAN Dataset

We tried to apply our method of image augmentation by chairs to DISCOMAN dataset [10]. As a first step, we took subsamples of 50,000 and 10,000 training images of DISCOMAN dataset. Moreover, the 10,000 images dataset was a subset of the 50,000 images dataset. After that, we first tried to augment a subset of the 10,000 images with synthetic chairs, and then improve them using a neural network. We also tried to extend the dataset with additional background images to compare the extension by useless images to the chairs augmentation. We chose the depth estimation and panoptic segmentation tasks as target neural networks.

While training the GAN in image post-processing, we cropped from image areas containing real and synthetic chairs and passed them as negative and positive examples to the GAN. Because if we submitted the whole picture to the input, then the neural network could not find a chair and was not trained.

Let us introduce the notation:*disc50k*—A set of 50,000 unmodified images from the DISCOMAN training set. After deleting invalid tracks 35,410 images remained.*disc10k*—A set of 10,000 unmodified images from the DISCOMAN training set, a subset of *disc50k*. After deleting invalid tracks 7125 images remained.*disc10k + aug (ours)*—A set of augmented by chairs images of *disc10k*. Total 50,970 images.*disc10k + aug-gan (ours)*—A set of images from *aug* post-processed by our GAN network described previously.

We first trained the neural network for the depth estimation task on different datasets and then evaluated the model on the DISCOMAN dataset. The results are in Table 5.

It can be seen that our pipeline improves the accuracy of depth estimation on the DISCOMAN dataset. However, it is still far from the desired values (*disc50k* as a reference value).

### 5.5. Panoptic Segmentation

To test our method on the scene parsing task, we trained the panoptic segmentation network proposed in [48] and compared the results of different post-processing approaches on the test dataset. The basis of their architecture is a simple Mask-RCNN network. They proposed to attach a new simple and fast additional semantic segmentation branch to the Feature Pyramid Network. For the branch of instance segmentation, they used a binary segmentation mask prediction for each candidate region of the detector.

Panoptic Quality (PQ) is used as a panoptic segmentation metric. PQ is a product of Recognition Quality (RQ) and Segmentation Quality (SQ). For each class, the predicted and ground truth segments are split into three sets: matched pairs of segments (TP—True Positives), unmatched predicted segments (FP—False Positives) and unmatched ground truth segments (FN—False Negatives).
PQ=SQ×RQ,
$$SQ=\frac{\sum_{(pred,gt)\subset TP}IOU\left(pred,gt\right)}{\left|TP\right|}$$
$$RQ=\frac{\left|TP\right|}{\left|TP\right|+\frac{1}{2}\left|FP\right|+\frac{1}{2}\left|FN\right|}$$

There are also two types of objects: Things—countable objects such as chairs, tables and nightstands, and Stuff—heterogeneous regions of similar texture or material such as a background. Panoptic segmentation examples are shown in the Figure 18.

#### 5.5.1. Results on NYUv2 Dataset

As can be seen from Table 4, the use of the NYUv2-aug image augmentation provides an improvement in the depth estimation task compared to using only the small original NYUv2-sm dataset. Post-processing with GAN did not improve the quality of depth estimation methods. As for the results of panoptical segmentation, presented in Table 6, we see that synthetic data, in general, did not give an increase in quality. Perhaps this is due to the fact that the synthetic data there most likely have some kind of artifacts or color inconsistencies. They are not visible to the eye, but the neural network learns to detect them, and not to extract semantic information to search for objects. Thus, the Recognition Quality decreases while the Segmentation Quality slightly increases.

#### 5.5.2. Results on DISCOMAN Dataset

Then, we trained the neural network for the panoptic segmentation task on different datasets and evaluated the models on the DISCOMAN dataset. We calculated the panoptic segmentation metrics. The results are in Table 7.

Let us extended the notation for DISCOMAN dataset:*disc50k*—A set of 50,000 unmodified images from the DISCOMAN training set. After deleting invalid tracks 35,410 images remained.*disc10k*—A set of 10,000 unmodified images from the DISCOMAN training set, a subset of *disc50k*. After deleting invalid tracks 7125 images remained.*disc10k + aug (ours)*—A set of augmented by chairs images of *disc10k*. Total 50,970 images.*disc10k + aug-gan (ours)*—A set of images from *aug* post-processed by our GAN network described previously.*disc50k-back*—A set of background images from *disc50k* that do not contain any objects. Total 3669 images.

As can be seen from the results, the introduction of synthetic chairs allowed us to improve the quality of panoptic segmentation and depth estimation. The PQ metric in the area with chairs increased from 35.0 to 36.2, and the total metric for all classes from 49.3 to 50.1, the RMS (root mean square error) of the depth estimation decreased from 0.4970 to 0.4369, that is an improvement. However, the quality still did not reach the quality when using a larger dataset. Augmentation of images with new backgrounds has a marginal impact on the neural network quality. The current method of image post-processing using GAN does not yet provide an increase in quality and, judging by the metrics, even slightly spoils it by marginal error.

## 6. Conclusions

We proposed a general pipeline aimed at improving of the accuracy of CNN-based methods via computer graphics and validate it on many fields and various datasets. However, it cannot be said that this increase occurs automatically by simply increasing the size of dataset using computer graphics or neural network synthesis techniques. Our experiments with DISCOMAN clearly shows that the quality still did not reach the “reference” quality when using a larger dataset.

At the same time for rare objects (traffic signs) we managed to increase the accuracy of the classifier from 0% upto 70–76%. This means that the proposed approach is valuable when we know the problems and specific properties of the dataset. To achieve a significant improvement in accuracy, it is important to be able to accurately model the distribution of specific objects in a specific dataset, i.e., a precise “sampling” by synthetic objects and their locations from the desired distribution. In our solution, this is achieved by the extendable features of the computer graphics engine, the scripting capabilities of our framework, and traditional and neural network image post-processing.

We also had succeed for the cars detector (mAP metric increased from 40.3% to 43.26%) due to realistic rendering of various cars and fine tuning of distribution parameters, which is possible due to the proposed pipeline. Our approach allows to investigate the influence of individual factors (Table 3), which is important for human-controllable distribution tuning. Since we trained the neural network on one dataset (KITTI) and evaluated the accuracy on another (Cityscapes), we can say that the proposed approach really improves the ability of the neural network to detect the objects we model.

Nevertheless, the main limiting factor of our approach is the high labor intensity for new areas and new datasets. It is like making several independent computer games having some general purpose or specific game engine. In our case, such an engine (our pipeline) is designed for sampling synthesized objects from a programmable distribution. However, the distribution (the game) itself must be tuned by creating 3D content, setting up materials, scripting distribution of parameters and placement logic. It usually takes us about a week (manpower) to create randomization script, tune all parameters, etc., to generate a new dataset.

We believe that reducing the labor intensity of tuning such distribution is a fundamental problem of the same complexity as reducing the labor intensity of the computer games or film production. For dataset generation some progress has already been made in this area [23,34].

## Figures and Tables

**Figure 1 sensors-22-02080-f001:**
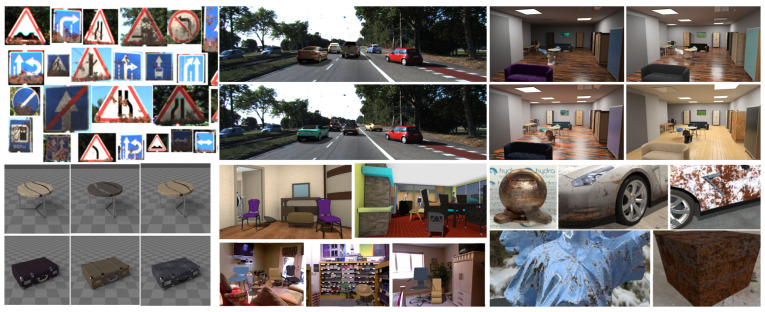
Examples of our applications. Road signs and cars were rendered as individual objects and then augmented to the KITTI dataset for further CNN training and testing (left and middle); realistic rendering of the interior, the 3D scenes are created randomizing of objects layout, material and lighting by our generator.

**Figure 2 sensors-22-02080-f002:**
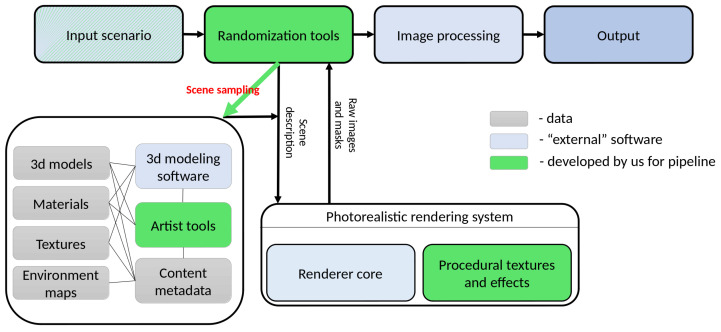
Proposed architecture of a synthetic data generation system. The green boxes represent the custom parts of the pipeline that we have developed in this work. The green arrow “Scene sampling” illustrates the process of scene rendering from a set of input random parameters that we discuss in Section 3.1.

**Figure 3 sensors-22-02080-f003:**
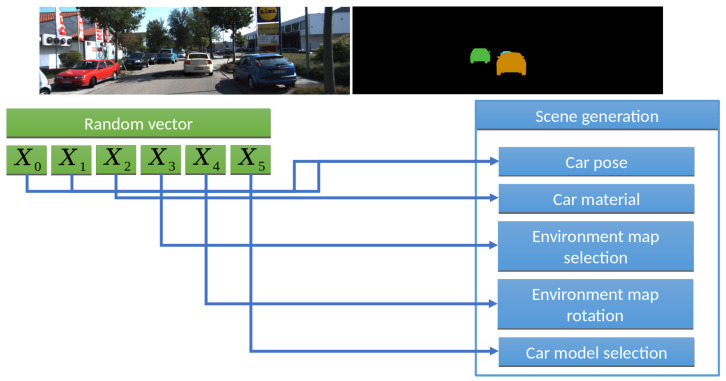
Example of assigning random numbers to scene parameters. The image result is demonstrated at the top left part of the figure. The output masks for synthesized objects are shown at the top right part of the figure.

**Figure 4 sensors-22-02080-f004:**
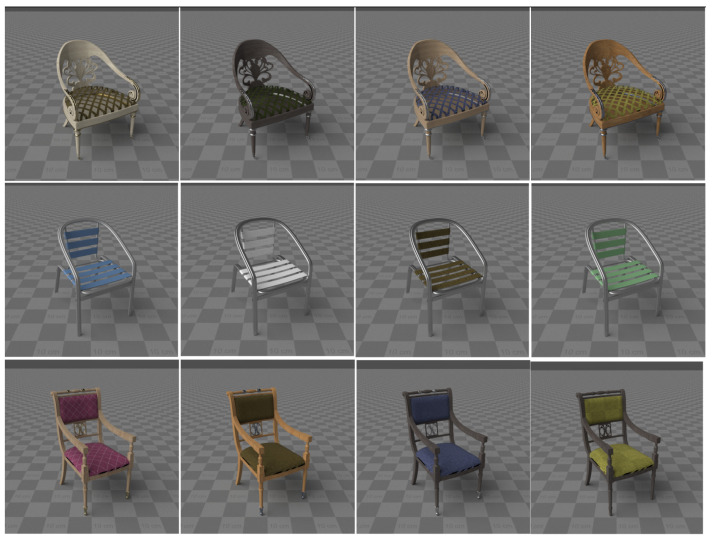
Examples of preview for randomization results of assigning random materials to the different 3D chair models.

**Figure 5 sensors-22-02080-f005:**
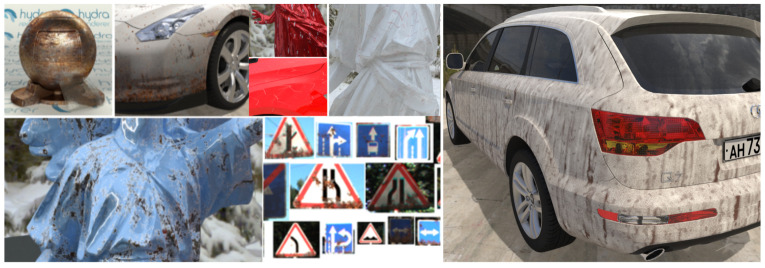
Examples of developed procedural textures.

**Figure 6 sensors-22-02080-f006:**
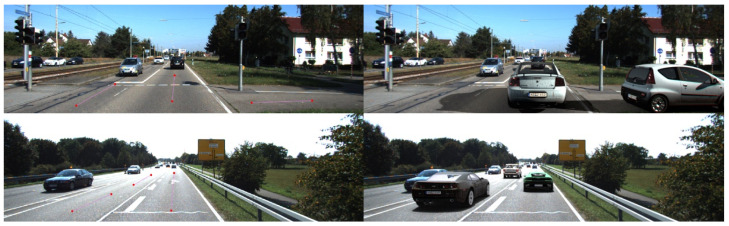
Marking trajectories on the original images (**left**) and augmenting image by rendered models at random positions on the trajectories (**right**).

**Figure 7 sensors-22-02080-f007:**
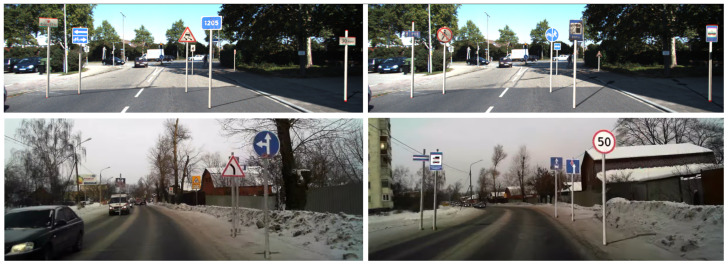
Examples of augmentation road signs in KITTI [13] (**top line**) and RTSD [37] (**bottom line**).

**Figure 8 sensors-22-02080-f008:**
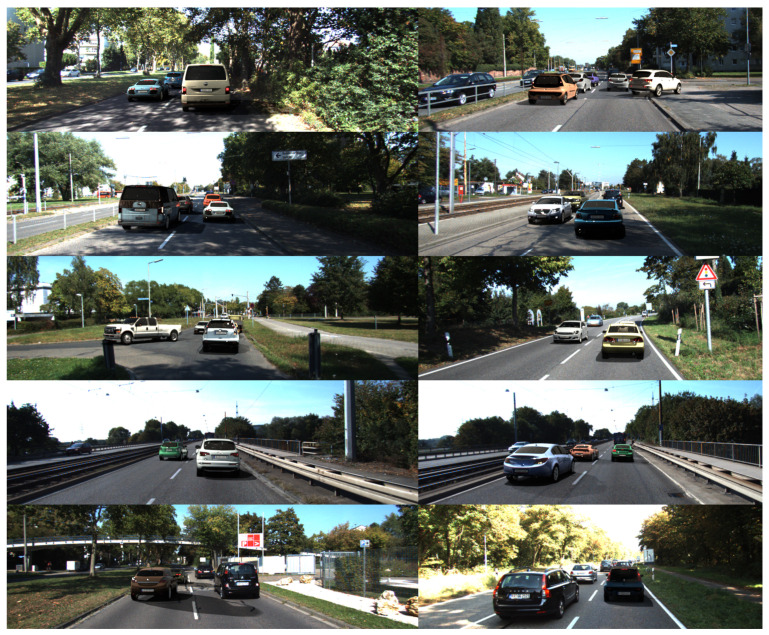
Some examples of augmented cars for KITTI dataset. All sub-images have equal meaning. They depict cars, some of which are synthetic and some are real.

**Figure 9 sensors-22-02080-f009:**
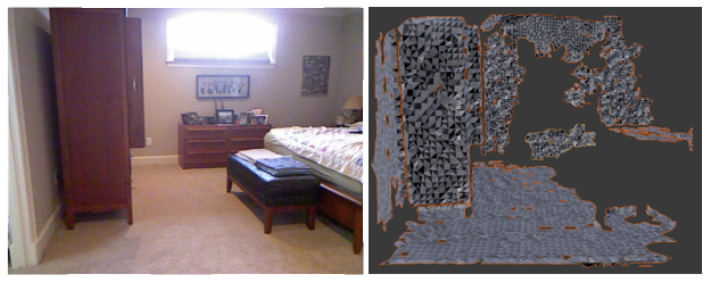
An example of plane reconstruction for an image from NYUv2 dataset [38].

**Figure 10 sensors-22-02080-f010:**
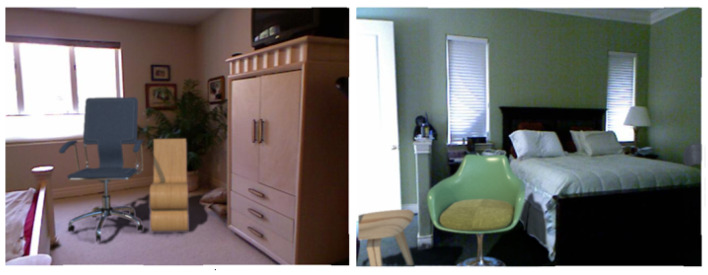
Example of embedding 3D objects in NYUv2 dataset [38].

**Figure 11 sensors-22-02080-f011:**
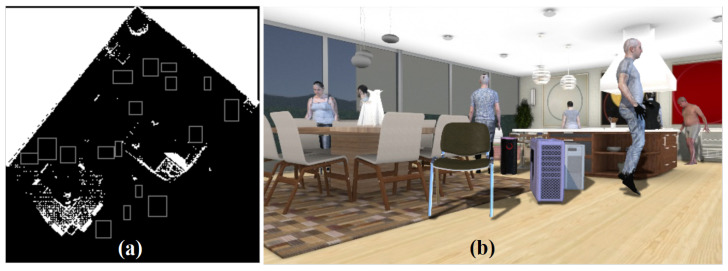
Example of the reconstructed occupancy map (**a**) and final image with embedded objects (**b**). Boxes show placeholders for the inserted objects.

**Figure 12 sensors-22-02080-f012:**
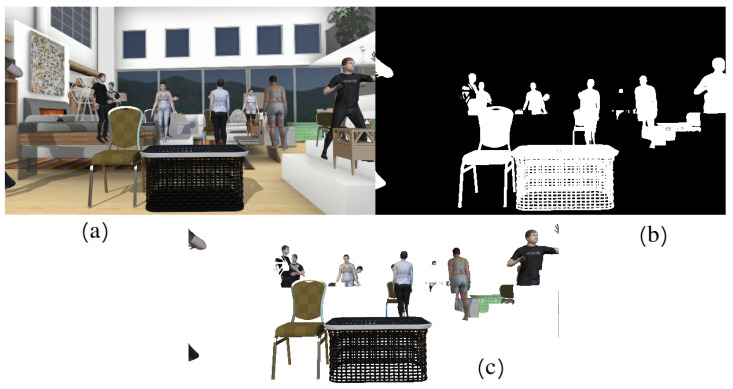
Rendered image (**a**), object mask (**b**) and objects cutout (**c**).

**Figure 13 sensors-22-02080-f013:**
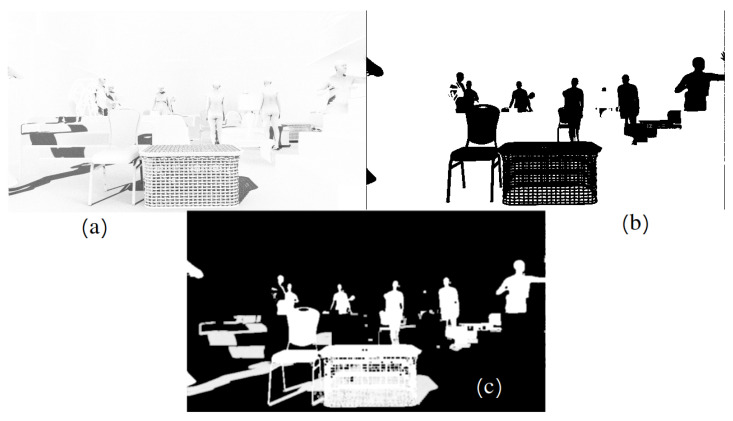
Shadow pass from GBuffer (**a**), shadow catcher mask (**b**), final shadow mask (**c**).

**Figure 14 sensors-22-02080-f014:**
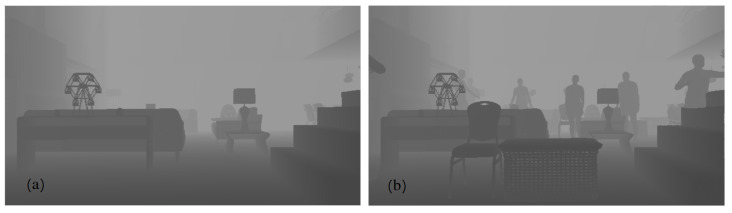
Depth buffer on the left (**a**) and augmented depth buffer on the right (**b**).

**Figure 15 sensors-22-02080-f015:**
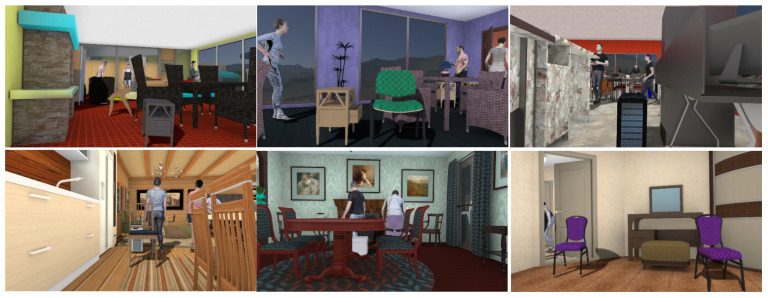
Examples of augmentation of the rendered models in DISCOMAN dataset.

**Figure 16 sensors-22-02080-f016:**
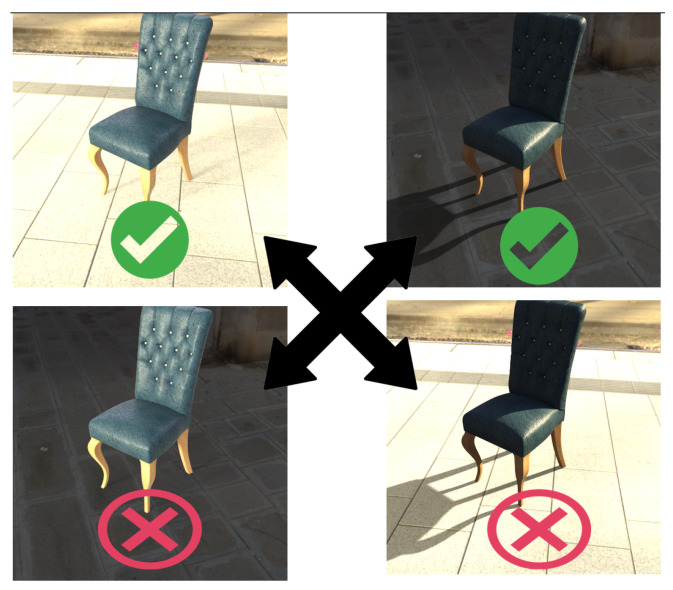
We created a training dataset to train lighting transfer through CycleGAN by generating pairs of augmented 3D models in an HDR environment. The top two images are correct because the models are illuminated by the environment they are placed in (augmented correctly). The two bottom images are obtained by replacing the environment of the background. We insert the model into one environment, but illuminate it with another one.

**Figure 17 sensors-22-02080-f017:**
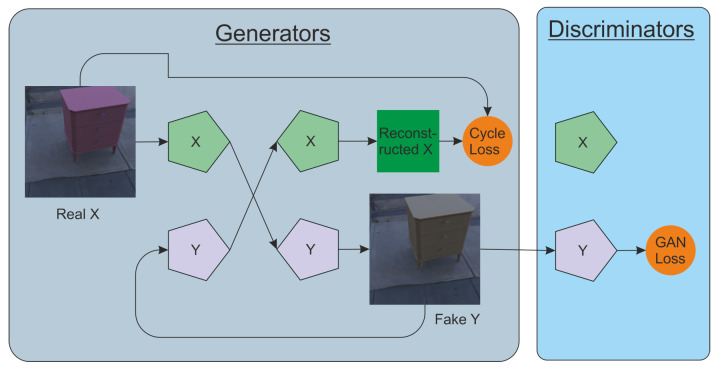
Scheme of CycleGAN modification used as the basic method of image post-processing.

**Figure 18 sensors-22-02080-f018:**
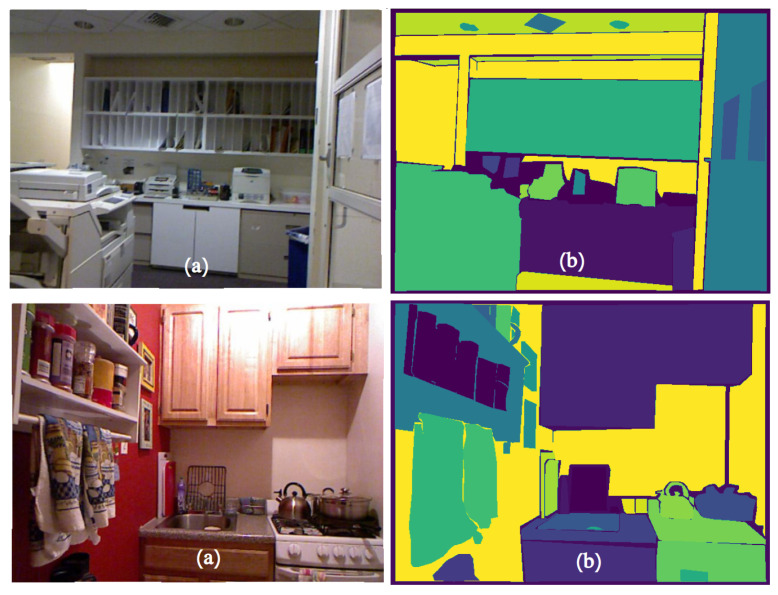
Panoptic segmentation of indoor scenes from the NYUv2 dataset examples. Real RGB images are on the left (**a**) and ground truth panoptic segmentations are on the right (**b**).

**Table 1 sensors-22-02080-t001:** Simple WideResNet classifier trained on a mixture of real and synthetic samples. Numbers in bold indicate the best result in the column.

	All, Accuracy	Rare, Recall	Frequent, Recall
RTSD	88.87	0.00	94.88
RTSD + CGI (ours)	92.102	70.16	93.59
RTSD + CGI-GAN (ours)	93.752	72.50	95.09
RTSD + Styled (ours)	**94.11**	**76.33**	**98.52**

**Table 2 sensors-22-02080-t002:** Accuracy of PVANet [45] detector on a mixture of a real and synthetic samples with and without classifier [43]. Ours approaches are “RTSD + CGI” and “RTSD + Styled”. Numbers in bold indicate the best result in the column.

	Without Classifier			With Classifier		
	All	Rare	Frequent	All	Rare	Frequent
RTSD	89.09	85.86	89.25	86.01	58.56	86.61
**RTSD + CGI (ours)**	88.56	85.72	88.72	83.84	48.51	85.15
RTSD + Inpaint	88.61	86.63	88.71	76.41	34.00	82.93
RTSD + Pasted	88.98	86.59	89.09	85.81	59.98	86.40
**RTSD + Styled (ours)**	89.01	**86.78**	89.13	85.39	64.20	86.13
RTSD + NN-additional	**89.17**	86.62	**89.31**	**86.16**	**64.96**	**86.70**

**Table 3 sensors-22-02080-t003:** mAP of car detector trained on different samples. Numbers in bold indicate the best result in the column.

Model	KITTI			Cityscapes		
	AP^0.5^	AP^0.7^	mAp	AP^0.5^	AP^0.7^	mAp
orig	74.62	49.31	40.30	41.00	24.24	23.09
aug-1 (ours)	**76.68**	50.99	**43.26**	46.64	28.08	24.57
aug-2 (ours)	76.04	50.99	42.32	42.09	28.09	23.74
aug-3 (ours)	75.56	50.44	41.77	46.07	**29.07**	24.70
aug-4 (ours)	75.59	50.94	41.81	46.11	27.71	**24.78**
aug-5 (ours)	76.30	50.08	41.99	44.61	26.98	23.63
aug-6 (ours)	74.81	**51.06**	42.14	46.89	28.71	24.68
aug-7 (ours)	74.48	51.01	41.54	45.26	26.29	23.26
aug-8 (ours)	75.38	50.69	41.88	47.08	27.61	24.15
aug-9 (ours)	76.12	50.98	42.45	**47.29**	27.40	24.64

**Table 4 sensors-22-02080-t004:** Neural network results for estimating depth maps on NYUv2 set and its augmentations. Numbers in bold indicate the best result in the column.

Training Data	Metrics					
	$a_{1}↑$	$a_{2}↑$	$a_{3}↑$	rel↓	rms↓	$\log_{10}↓$
small	0.7373	**0.9395**	0.9839	0.1804	0.5807	0.0735
small + aug (ours)	**0.7409**	0.9336	0.9818	0.1867	**0.5706**	**0.0733**
small + aug-gan (ours)	0.7315	0.9376	**0.9849**	**0.1780**	0.5932	0.0744

**Table 5 sensors-22-02080-t005:** Comparison of depth estimation metrics, when trained on various original and post-processed combinations of DISCOMAN data. Numbers in bold indicate the best result in the column.

Training Data	Metrics on Discoman					
	$a_{1}↑$	$a_{2}↑$	$a_{3}↑$	rel↓	rms↓	$\log_{10}↓$
disc10k	0.8277	0.9261	0.9547	12.3888	0.4970	0.0864
disc10k + aug (ours)	0.8385	0.9262	0.9526	**10.2756**	**0.4369**	0.0813
disc10k + aug-gan (ours)	**0.8432**	**0.9395**	**0.9706**	13.0993	0.4839	**0.0694**
disc50k (reference)	0.8942	0.9632	0.9817	10.5944	0.4040	0.0539

**Table 6 sensors-22-02080-t006:** Neural network results for panoptic segmentation on NYUv2 dataset and its augmentations. Numbers in bold indicate the best result in the column.

Training Data	All			Things			Stuff			Chairs		
	PQ	SQ	RQ	PQ	SQ	RQ	PQ	SQ	RQ	PQ	SQ	RQ
small	**34.1**	75.9	**43.8**	**28.5**	74.8	**37.7**	**69.2**	83.1	**83.0**	**36.7**	73.6	**49.9**
small + aug (ours)	32.5	76.0	41.7	26.8	74.8	35.3	68.8	**83.5**	82.1	36.5	**73.8**	49.4
small + aug-gan (ours)	32.1	**76.3**	41.2	26.4	**75.3**	34.7	68.3	82.9	82.1	36.1	73.2	49.3

**Table 7 sensors-22-02080-t007:** Comparison of the panoptic segmentation methods trained on various original and post-processed combinations of DISCOMAN data. Numbers in bold indicate the best result in the column.

Model Trained On	All			Things			Stuff			Chairs		
	PQ	SQ	RQ	PQ	SQ	RQ	PQ	SQ	RQ	PQ	SQ	RQ
disc10k	49.3	79.7	58.3	45.6	77.8	54.9	62.7	87.7	**70.4**	35.0	75.8	46.1
disc10k + aug (ours)	**50.1**	**80.2**	**59.1**	**46.4**	77.8	**55.7**	**62.5**	**88.4**	70.0	**36.2**	76.6	47.2
disc10k + aug-gan (ours)	49.1	79.7	58.0	45.7	**77.9**	54.8	61.2	86.8	69.3	35.9	**77.0**	**46.6**
disc10k + disc50k-back	48.5	78.6	57.2	44.9	**77.9**	53.4	62.1	87.3	69.5	35.2	76.1	46.2
disc50k (reference)	54.1	81.5	62.9	50.7	79.4	59.9	65.1	88.4	72.8	39.7	77.8	51.0

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
