# Peer review of "Image Synthesis Pipeline for CNN-Based Sensing Systems"

_sensors, 2022, doi:10.3390/s22062080_

Round 1

Reviewer 1 Report

The paper proposed a pipeline that using synthetic/generated/rendered 2D images for several computer vision task, namely object detection, segmentation and depth estimation. However classification task is not included in the experiments. In general the paper provided lots of implementation and engineering details however the English and style of the paper are not suitable for publication. The idea that used in the paper is not uncommon to be used as data augmentation to improve the performance of machine learning models. Contributions seems not clear for me. There are substantial improvement that is needed.

1. The title doesn’t make sense. It’s not grammatically right.

2. The paper tend to overstate the contribution of the work. In the abstract and multiple sentences in the paper, the authors are claiming they solved the problem of data quantity and quality issues from training CNNs. This is not the case if we see the results reported in the experiment part. For example in section 2.7 the authors mentioned improve accuracy of computer vision algorithms. The application domain is too broad which is not backed by the experiments as well.

3. There are couple of terminology that should avoid using in scientific paper such as AI training experiments.

4. The paper is badly written, and I suppose without proof reading. There are couple of places the English is not comprehensive such as 

a) The efficiency of this type of sensors largely depends on the quality of neural network training. Moreover, different training data may be required to successfully solve different problems.

b) The result sounds like this: it doesn’t matter how well we model the road signs themselves. 

        c) In section 2.5. “Marking training samples” is normally referred as labelling or annotating.

There are many many sentences like this which needs to be fixed.

5. The authors are putting lots of emphasis on reproducibility. However I am not sure if it the right term to use. And why is matching data distribution of generated and real ones has to be “exactly the same position, colors, etc. of rendered cars with and without dirt”? I would like to see more experiment to back this claim.

6. Figure 2 need to be updated. Why is the scene sampling arrow green?

7. In the example of figure 3, how is the background generated are they also generated from 3d models? If the background is taken from the original dataset, have you considered using lidar sensor data? I see that you are using vehicle trajectories to compute a matrix that can project vehicle to real world coordinate(section 4.1), when you place new vehicles is there any randomness when you choose where to place the vehicle or all synthetic vehicles has to put on the previous trajectories?

8. Section 5.2 needs to be rewritten. In table 1 what is the metric that is reported? Lots of methods that mentioned in the table are either not referenced or mentioned in the previous sections.

9. In Table 2, instead of demonstrate the performance of disabling features, I think it more important to demonstrate the performance of using certain features. It is not clear for me during training do you still using training data from KITTI and Cityscape.

10. In general most of the experiment result reported in the paper are only marginally better than just using data without augmentation. Only rare road sign and cars detector has meaningful improvement. I suggest the authors to put extra effort on analysing what makes the improvement in these 2 cases which I think it’s more interesting for the community. A more comprehensive ablation study is needed for those 2 cases.

It is an interesting topic in machine learning regarding using synthetic images to match existing training data distribution. I appreciate the amount of engineering effort demonstrate in the paper, however I am not convinced by the results demonstrated. The experiments could be better analyzed and presented.

Reviewer 2 Report

This is an interesting study to present a pipeline for image dataset augmentation by synthesis with computer graphics and generative neural networks approaches. The proposed approach is evaluated and demonstrated in a few benchmark datasets in the computer vision domain.

Some details of the proposed system can be further clarified.
1.  Figure 3 presents various key parameters that can be controlled to generate data. What would be the range for these values?
2. I have some minor concerns regarding the 'accuracy improvement' in the paper title. It is a too general term. It might be rephrased to be 'Image synthesis pipeline for CNN-based sensing systems' or other names.
3. Line 563: Please clarify what does '0% accuracy' means in the statement 'increase the accuracy of the classifier for the rare traffic signs from 0% to 72.5% [41].'
4. Line 125: 'iss usually limited' -> 'is usually limited'.
5. Since manual tuning and image polishing are involved in the proposed pipeline, it would be good to comment on the manpower (e.g., how many hours are needed) needed for the dataset synthesized.

Round 2

Reviewer 1 Report

I am glad to see the authors took my suggestions and made lots of changes. I think the paper is in a much better position now. After checking the updated manuscript, there is only a few issues I would like to raise.

1.   I understand that the proposed pipeline is to create datasets such that the new dataset has some desirable distributions that the original dataset doesn’t. In the response regarding the necessity of using exactly the same position, colours, etc. of rendered cars with and without dirt. My concern would be the impact of the prior distribution of objects in the background. For instance in KITTI or NYUv2 datasets, synthetic objects are rendered and directly put into the original images. Let’s use the example of impact of dirt on the cars. If the original dataset are full of cars with dirt how could you make sure the newly generated dataset are actually uniformly distributed of cars with and without dirt. Since you  don’t know the real distribution of the dataset you are trying to augment. In other words, all the augmented datasets are always the distribution of prior + synthetic distribution, so how much weight the prior can bias the experiments really needs to be discussed somewhere in the paper.

2. I think the authors needs to added some works of conditional GAN and VAE, since I think what you are trying to do is something similar, which is to model dataset distribution and generate new ones with certain distribution/conditions. Whether the conditional GAN and/or VAE can be easily trained is another problem, however they seems to be faster than proposed approach.
